# Awareness, Knowledge, Attitude and Preventive Practice of Leptospirosis Among Healthy Malaysian and Non-Malaysian Wet Market Workers in Selected Urban Areas in Selangor, Malaysia

**DOI:** 10.3390/ijerph17041346

**Published:** 2020-02-19

**Authors:** Suhailah Samsudin, Sakinah N.S. Saudi, Norbaya S. Masri, Nur Raihana Ithnin, Jamaluddin T.Z.M.T, Rukman A. Hamat, Zahiruddin W.M. Wan Mohd, Mohd S. Nazri, Sukeri Surianti, Aziah B. Daud, Munirah N. Abdullah, Nozmi Noramira, Malina Osman

**Affiliations:** 1Department of Medical Microbiology and Parasitology, Faculty of Medicine and Health Sciences, Universiti Putra Malaysia, Serdang 43400, Selangor, Malaysia; suhailah.sam@gmail.com (S.S.); sitinorbaya@upm.edu.my (N.S.M.); raihana@upm.edu.my (N.R.I.); rukman@upm.edu.my (R.A.H.); nurulmunirahabdullah@gmail.com (M.N.A.);; 2Department of Community Medicine, School of Medical Sciences, Universiti Sains Malaysia, Kubang Kerian, Kelantan 16150, Malaysiadrnazri@usm.my (M.S.N.); surianti@usm.my (S.S.); aziahkb@usm.my (A.B.D.)

**Keywords:** knowledge, attitude, preventive practice, leptospirosis, urban area, prevention awareness

## Abstract

Leptospirosis has been reported as an endemic in most tropical countries. Among high risk occupations, leptospirosis includes workers in agriculture and domestic animal industries. Environmental hygiene in the wet market has established a link between the presence of rodents with probability of leptospirosis infection. This study was aimed to compare the level of knowledge, attitude and preventive practice against leptospirosis between healthy Malaysian and non-Malaysian wet market workers in selected wet markets in urban areas of Selangor. A cross-sectional study in the determined area was conducted with the participation of 147 respondents. The respondents were randomly chosen from the list provided by the state agency that regulates these markets. A self-administered bilingual validated questionnaire (English and Bahasa Melayu) was distributed to the selected respondents. There were 68 (48.3%) Malaysian respondents and 79 (53.7%) non-Malaysian respondents. The majority of them were males, who attained formal education and were less than 40 years old. Meanwhile, the respondents earned less than RM3000. Among the Malaysian respondents, 80.9% were aware of leptospirosis as compared to 17.7% of the non-Malaysian colleagues (*p* < 0.05). All items of knowledge showed that the Malaysian respondents scored higher as compared to non-Malaysian respondents. On attitude towards infection prevention, most Malaysian respondents had a positive attitude, while most non-Malaysian respondents had undecided perception on the majority of crucial attitude items. In practicing preventive measures, there was a marked significant difference in proportion between Malaysian and non-Malaysian respondents for items on “Specific Protection and Isolation at Source.” There was a significant gap on knowledge, attitude and preventive practice among Malaysian workers as compared to non-Malaysian workers. Therefore, it was highly recommended the health promotion implementation should also provide specific focus on non-Malaysian workers.

## 1. Introduction

Leptospirosis is an infectious disease that affects humans and animals. It is considered one of the world’s most widespread re-emerging zoonotic diseases [1,2,3]. The documentation of high leptospirosis prevalence in humans is of great public health concern, particularly in Malaysia [1,4]. Leptospirosis is directly or indirectly transmitted to humans [5] through contact with infected animal urine in water, soil, or other materials [6,7]. Among animals that are most likely to spread the disease are rodents (20%), marsupials (35%) and bats (35%) [8].

Wet markets are venues where the transaction of raw food materials like fresh meat, fish, vegetables or fruits takes place. All these activities provide a conducive environment which favours the presence of rodents [9,10]. Recent studies also showed that pathogens of *Leptospira* sp were present in trapped rodents and environmental samples in wet market areas. Poor environmental settings of wet markets are known as sources for certain infections and have attracted certain rodent species [10]. The increasing number of rodents in these areas is mainly due to presence of food remnants, poor sanitation and improper waste disposal [11].

Wet market workers may pose a risk to acquire leptospirosis through exposure to *Leptospira* in contaminated wet market environments. Those who are working in wet markets, such as slaughterhouse workers, veterinarians, animal caretakers, and fish workers are listed as high-risk occupations associated with leptospirosis [12].

The Department of Statistics Malaysia had documented that in 2018 there was 10.8% of international migration to Malaysia [13]. One of the major international migrations in Malaysia was among the refugee community. Malaysia has been accepting refugees from a humanitarian point of view, and the country did not sign the Refugee Convention [14,15]. Therefore, its role remains confined to receive refugees based on a humanitarian point of view without available clear lines that define and regulate their lives, issues, rights and demands in the country [14]. Attention is given only to some groups that have received United Nations High Commissions for Refugees (UNHCR) cards, but this did little to improve their access to various services, such as education and health [16,17]. A major setting where non-Malaysians are working include services sectors like plantations, restaurants and wet markets [18].

As awareness of leptospirosis among wet market workers is not prevalent, a study on knowledge, attitude and preventive practice of leptospirosis among market workers was conducted at selected wet markets in urban areas of Selangor, Malaysia. Based on field observation, which identified that the proportion of non-Malaysian workers was higher as compared to Malaysian workers, this study would provide comparative data between Malaysians and non-Malaysians. The study data would provide baseline information on the level of knowledge, attitude and practice towards leptospirosis prevention among wet market workers. Therefore, an appropriate interventional study can be established accordingly.

## 2. Materials and Methods

### 2.1. Study Design, Sampling Population and Sample Size

A cross-sectional study was conducted in selected wet markets. There were eight wet markets listed by the relevant Malaysian authorities. Four wet markets were selected randomly. The selected wet markets were Pasar Besar Kajang, Pasar Seksyen 1 (Bandar Baru Bangi), Pasar Borong Selangor, and Pasar Taman Sri Serdang.

Eligible respondents from the targeted population, i.e., wet market workers, were selected through cluster sampling. In this study, every individual, either Malaysian or Non-Malaysian (including those who hold a UNHCR card), who worked as a wet market worker in the selected study areas for more than six months were included. Meanwhile, those who were below 18 years old and had a history of recent fevers, and those who were illiterate and did not understand Bahasa Melayu or English were excluded. A database of wet market workers was created based on information provided by the Malaysian authorities. The sample size was determined based on research design and objectives. In this study, the calculated sample size was 109 [19].

### 2.2. Instrument

A self-administered questionnaire that was modified and adapted from previous studies [13,14,15] in dual language (English and Bahasa Melayu) was distributed among workers who agreed to participate in the study. Respondents who were not familiar with the formal written language were assisted by trained language translators. The questionnaire consisted of five main sections which covered demographic and socioeconomic information, knowledge about leptospirosis, attitude towards leptospirosis prevention and preventive leptospirosis practices. Content validation was carried out through a discussion with content experts and cross-referenced with documented scientific materials. Construct validation was done through item analysis in which questions with low reliability were deleted. Assessment of reliability was done through internal consistency reliability analysis. Cronbach’s alpha values for scope of knowledge, attitude and practice in the questionnaires were 0.71, 0.76 and 0.70, respectively. The questionnaire was pre-tested among food workers at similar settings for more than 10% of the sample size to ensure that the questions were understood.

The knowledge section consisted of 22 questions which included respondents’ knowledge of causative agents, modes of transmission, signs, symptoms and complications, risk factors, treatments, diagnosis and prevention. Questions about knowledge were designed to be answered in the correct or incorrect format.

The attitude section comprised of 13 questions related to the individual’s attitude towards leptospirosis prevention. These included risk of infection due to occupational exposure, hobby, and environmental setting, attitude during seeking treatment after being infected by leptospirosis, the individual’s perception on the disease progress if tested as positive, including complication and death, attitude on prevention of leptospirosis from perspectives of personal hygiene and environmental cleanliness. Questions on attitude were assessed using a Likert scale (“strongly agree”/“agree”/“neutral”/“not agree”/“strongly not agree”). For positive attitude items, scores of “5”, “4”, “3”, “2”, and “1” for “strongly agree”, “agree”, “ not sure”, “not agree”, and “strongly not agree” were awarded, respectively, and reversed for negative attitude items.

The preventive practice section consisted of 12 questions. This section had questions on personal hygiene (“washing hands with water and soap either after using the toilet or before eating”), environmental hygiene (“practice on trash disposal, cleanliness of equipment used, and premises”), specific protection (“wearing of boots, apron, and covering of each wound with plaster neatly”), isolation (“how respondents stored their stuff to prevent rat droppings”), and elimination (“methods to eradicate rats by using poison, trap, and cleaning programmes with the Malaysian authority”). Questions about practice were answered as “yes” or “no”.

### 2.3. Data Analysis

Statistical methods for analysing data in this study used IBM SPSS Statistics Version 25.0. Analysis was done according to specific objectives. In this study the level of statistical significance was set at *p* < 0.05.

### 2.4. Study Ethics

Ethical approval was obtained from the Medical Research Ethics Committee, Universiti Putra Malaysia (JKEUPM), and permission from the Kajang Municipal Council (MPKj) and Subang Jaya Municipal Council (MPSJ) prior to the study.

## 3. Results

### 3.1. Distribution of Demographic and Socioeconomic Indicators

A total of 147 respondents participated in this study, of which 68 (48.3%) were Malaysian workers and 79 (53.7%) were non-Malaysian workers. The mean (SD) of age and monthly income of Malaysian respondents were 40.8 (14.3) and RM 2255 (1684), while for the non-Malaysian respondents, their mean (SD) age and monthly income were 33.9 (10.5) and RM 1298 (655.6), respectively.

For Malaysian workers, the majority were more than 40 years old (57.4%), male (63.2%), and attained formal education (88.2%) with incomes less than RM 3000 (83.8%). Most of them were involved in retail businesses and selling vegetables and fruits (64.7%) (Table 1).

Most non-Malaysian workers were Rohingya and Indonesians, and were working in the market for more than six months. Most of them were less than 40 years old (72.2%), male (78.5%) and had attended formal education (63.3%) with low incomes (97.5%). More non-Malaysian workers were involved in selling fish and seafood (25.3%) as compared to Malaysian workers (5.9%) (Table 1).

### 3.2. Description of Knowledge of Leptospirosis

On knowledge about leptospirosis, it was found that 53.1% of respondents did not know about rat urine disease (leptospirosis). In comparison, between those who do not know about leptospirosis, 82.3% of them were non-Malaysian workers and 19.1% were Malaysians. There was a significant difference in the proportions between both groups (*p* < 0.001) (no table).

Generally, the Malaysian workers scored higher marks as compared to non-Malaysian workers for the main questions under the knowledge domain. Most of them were aware that leptospirosis is caused by bacteria (67.6%), transmitted through contaminated food (64.7%), presented as muscle pain (50%), causes complications in lungs and kidneys (54.4%) and causes death (73.5%). The majority of Malaysian workers were also aware that participating in recreational activities (55.9%), eating street food (60.3%) and living near flood areas (60.3%) are risk factors for leptospirosis. Meanwhile most of them also knew about disease prevention. The Malaysian respondents were also aware of infection detected through blood screening (66.2%), and they knew that it was prevented through cleanliness in the house/workplace (79.4%), proper storing of food/goods (61.8%), avoiding walking without shoes (69.1%) and installing rat traps (57.4%).

The distribution of scores among non-Malaysian workers who answered correctly showed a unanimous trend in which most of them scored low (less than 20%) for all of the questions under the knowledge domain.

This study also found that for all knowledge domain scopes, the non-Malaysian workers showed a lower score as compared to Malaysian workers. They scored lower for questions on items of causative agents, modes of transmission, signs and symptoms, risk factors and for diagnosis, treatment and prevention (Table 2). Except for questions on presence of jaundice under items of signs, symptoms and complications, all other questions showed significant differences of scores who answered correctly between Malaysian and non-Malaysian workers (*p* < 0.001).

### 3.3. Description of Attitude Items Toward Prevention of Leptospirosis

There were 13 statements in attitude items which stated either a positive attitude or a negative attitude toward leptospirosis prevention. The majority of respondents from both groups (85.3% Malaysians and 64.6% non-Malaysians) unanimously agreed and strongly agreed that eating food contaminated with rat excretion poses a risk of infection to susceptible individuals. Most of the respondents (91.2% Malaysians and 63.3% non-Malaysians) strongly agreed and agreed that the presence of rats in their house/workplace imposed higher a risk for them to get infected. They also agreed (75% Malaysians and 59.4% non-Malaysians) that uncovered bins attract rats to the area. On wading in the flood, distribution of perception among Malaysian respondents was inconclusive, while most of the non-Malaysian respondents disagreed or were undetermined. On whether their occupation may expose them to contract the infection, the majority of Malaysian respondents strongly agreed or agreed (55.8%), while most non-Malaysians were undetermined (46.8%).

Meanwhile most respondents agreed that specific hobbies may expose individuals to the infection, the majority of non-Malaysian workers (50.6%) had undecided perceptions on this item. About one third of them were also found to be undecided whether early treatment is important to prevent death and serious complications; and most non-Malaysian workers were undecided on the risk of severe infection.

The majority of Malaysian workers (more than 70%) agreed that knowing the disease and utilising personal protective equipment (PPE) are important to prevent infection, but for the non-Malaysian workers, about a third of them were undecided about both items.

A similar trend was also observed for questions about cleaning programmes by the Malaysian authorities (Table 3).

### 3.4. Description of Practice Items Toward Prevention of Leptospirosis

Both Malaysian workers and non-Malaysian workers had good hand hygiene practices. The majority of them, i.e., 97.1% of Malaysian workers and 94.9% non-Malaysian workers would “wash their hands with water and soap before and after using the toilet”, while 94.1% of Malaysian workers and 94.9% non-Malaysian workers would “wash their hands with water and soap before and after preparing food or work”. There was no statistical difference for both groups of workers.

Regarding environmental hygiene, both groups mostly practiced the cleaning activities, such as “they washed their business utensils before and after trade every day” and “washing business site every day and appropriately managing the trash into bins provided.” The percentage of those who practised these among non-Malaysian workers was relatively lower as compared to hand hygiene (Table 4).

On utilising personal protective equipment (PPE), the percentage of Malaysian workers who used shoes/boots was higher (76.5%) as compared to non-Malaysian workers (54.4%). The difference in proportions was statistically significant (*p* < 0.05). Similarly, wearing an apron and covering a wound were appropriate. It was found that 85.3% of Malaysian workers covered their wounds with the appropriate cover (plaster) as compared to 62.0% non-Malaysian workers (*p* < 0.05).

As for isolation items, it was found that approximately 60% of Malaysian and non-Malaysian workers stored their items in appropriate containers beyond business hours. The proportion was almost the same and statistically there was no significant difference.

In terms of eradication at the source, the proportion of Malaysian workers who eradicated at the source was higher as compared to non-Malaysian workers. It was found that about 41.2% of Malaysian workers used rat poison as compared to 22.8% of non-Malaysian workers (*p* = 0.016). About 48.5% of Malaysian workers used rat traps to reduce the rat population as compared to about 19.0% of non-Malaysian workers (*p* = 0.0001). It was revealed that the majority of Malaysian workers (70.6%) did cleaning at the market sites as compared to 48.1% of non-Malaysian workers (*p* = 0.006) (Table 4).

## 4. Discussion

This study provided the first assessment on knowledge, attitude, and practice of leptospirosis comparisons among Malaysian and non-Malaysian workers in Selangor wet markets.

### 4.1. Distribution of Demographic and Socioeconomic Indicators

This study found that the majority of Malaysian and non-Malaysian respondents were less than 40 years old, male, had formal education and earned monthly incomes of less than RM3000. Incomes less than RM3000 among non-Malaysians were markedly high (97.5%) with their mean (SD) at RM 1298 (655.6). Sakinah et al. [20] reported that the majority of the respondents were less than 50 years old, which was quite similar with the study findings. Furthermore, De Araujo et al. [21] and Nozmi et al. [22] also found that the majority of respondents were males [15,16]. Most of the workers received formal education as similarly reported in [22]. Besides, the findings from this study were similar to Sakinah et al. [20] and Nozmi et al. [22] who found that the majority of the workers earned a monthly income of less than RM3000. However, there was a difference between Malaysian and non-Malaysian workers in types of business. Most Malaysian workers were directly involved as a leader in retail business and selling vegetables and fruits, while non-Malaysian workers were commonly assistants in selling fish and seafood. It was probably because there was a limit on obtaining business licenses for non-Malaysian workers as compared to Malaysian workers.

### 4.2. Distribution of Knowledge Regarding Leptospirosis Among Market Workers

The majority of workers, especially non-Malaysian workers, revealed that 82.3% of them did not know about leptospirosis as compared to Malaysian workers. A significant difference in knowledge regarding knowledge on leptospirosis between Malaysian and non-Malaysians (*p* < 0.001) showed that the non-Malaysian workers were probably never exposed to information regarding leptospirosis even though they had worked for more than six months in the country. These results were in accordance with [23,24], which indicated that non-Malaysians or non-citizens had more health disadvantages in the country destination, mainly because of low accessibility to health education programmes.

The majority of Malaysian workers answered incorrectly on leptospirosis as a zoonotic disease and mainly did not know that “leptospirosis can be transmitted via cuts and wounds on the body”. It is an important domain in assessing respondents who understand the disease because if they have knowledge about how transmission can occur, they will be able to prevent contracting the disease. Besides, most Malaysian workers correctly answered that leptospirosis can cause death, and they were able to relate a few risk factors, such as recreational activities, eating street food and living near flood areas with the disease. Few previous studies reported that recreational activities [25], eating contaminated food [26] and flooding [27] were significant risk factors in spreading leptospirosis to humans.

A lower score for all scopes under the knowledge domain was found among non-Malaysian workers in this study. The result showed significant differences between Malaysian and non-Malaysian workers (*p* < 0.001). One of the possible explanations for the differences in levels of knowledge between both groups is the language barrier [23]. Not all non-Malaysian workers were fluent in Bahasa Melayu or English; and most of them were refugees who were granted with special cards under the United Nations High Commissions for Refugees (UNHCR). The lack of understanding of questions and limitations of communication could be explanations for the low percentage of correctly answered questions by non-Malaysian workers.

### 4.3. Distribution of Attitude Regarding Prevention of Leptospirosis Among Market Workers

The study found that the majority of market workers had positive attitudes toward leptospirosis. However, there were a few domains which showed contrasting opinions between Malaysian and non-Malaysian workers.

Most Malaysian workers disagreed with the statement “wading in the flood does not pose a risk of leptospirosis” while non-Malaysian workers chose “undecided” on the statement. This showed a gap for this domain among Malaysian workers because the result showed that they had good knowledge of relating floods as a risk factor of leptospirosis, but their attitudes were differently reported. Previous research [20,26,27,28] found an association between infection of *Leptospira* and contact with floodwater. Therefore, if the workers had positive attitudes regarding flood exposure and avoided contact with flood water, cases of leptospirosis could be reduced.

This study showed that Malaysian workers had positive attitudes toward the statement of “working as market workers may expose them to leptospirosis” as compared to non-Malaysian workers. The transmission of leptospirosis among market workers may occur through exposure while taking out the trash and poor market sanitation [29,30]. Previous research [31,32] found a positive relation in the transmission of *Leptospira* to the market environment by rats. Therefore, it is an important attitude to be nurtured, especially to non-Malaysian workers for the prevention of leptospirosis.

Few hobbies or outdoor activities [20,21,33,34] may increase the risk of getting leptospirosis, yet the majority of non-Malaysian workers had a neutral opinion related to these risk factors. This may be explained by the finding by Farkas [35] who found that most non-Malaysians in their study setting (81%) perceived that they had to work hard to earn a better life, which then allowed them to have limited time for recreation or have any hobby. This may also be a similar trend in most non-Malaysian worker groups, as they had to work hard to earn a living. This also explained why they were less aware of the link of leptospirosis with specific recreational activities.

Most Malaysian and non-Malaysian respondents agreed that knowing about the disease could help in its prevention. They also agreed that utilising personal protective equipment during cleaning programmes and cooperation with Malaysian authorities would be among the recommended preventive measures. The findings were similar with S Saudi et al. [36] who found that the respondents also had similar attitude patterns on all items.

### 4.4. Distribution of Practice Regarding Prevention of Leptospirosis Among Market Workers

On preventive practices against leptospirosis, both Malaysian and non-Malaysian workers practised appropriate measures for personal and environmental hygiene (Table 4). Practising appropriate personal hygiene, such as washing hands, is crucial in leptospirosis prevention. More than 90% of respondents had good hygiene practice by washing hands with water and soap, before and after going to the toilet, as well as before and after preparing food. This finding was supported by [23,25,37] which recognised the importance of personal hygiene among workers.

The majority of respondents practiced appropriate environmental hygiene. Most of them cleaned their business sites, appropriately washing the utensils and ensured that all trashes were appropriately managed. Allwood et al. [38] reported that good sanitation prevented leptospirosis. The findings of this study concurred with de Araujo et al. [21] and Quina et al. [39], whereby most workers had good practice in preventing the infection.

On the item of specific protection, it was found that the percentage of respondents who had utilised personal protective equipment (PPE) among Malaysian workers was higher as compared to non-Malaysian workers (*p* < 0.05) (Table 4). Cediel et al. [23] also reported a similar finding, in which they found that more Malaysian workers used personal protective equipment as compared to non-Malaysian workers. Issues related to specific protection were linked to policy, prioritisation and financial resources [20]. The fact that non-Malaysian workers were struggling for their living perhaps led to the situation in which most of them did not utilise the specific protection.

In an occupational setting, elimination or eradication at the source was the most effective control measure. This was shown by Suhailah et al. [29] for the prevention of leptospirosis. The findings of this study indicated that there was a significant difference or proportion between Malaysian and non-Malaysian workers in all items of this domain (*p* < 0.05).

Getting rid of rats in the environment helps to reduce the probability of *Leptospira* maintainance in the environment. In this study, it was found that less than 50% of Malaysian and non-Malaysian workers used specific poison to eradicate the rats. The percentage was markedly low among non-Malaysian workers (less than 20%). (Table 4). This was supported by a previous report by Wong et al. [40] who had documented a significant relation between race and disease prevention.

On community engagement activity, it was found that less than half of the non-Malaysian respondents had participated in cleaning activities together with the Malaysian authority. Real situations for both groups of workers may be significantly different due to citizenship status and motivation for their life struggle, respectively. Chang et al. [41] documented on occupational health issues among non-Malaysian workers, detailing their plights while at the same time they had to work hard to earn a living. Limitation to access health facilities and difficulty to secure permanent jobs allowed non-Malaysian workers to focus more on their job scope rather than on handling issues related to health or communities. This result may explain why Malaysian workers are more active in preventing diseases such as leptospirosis.

The study limitation was that it was impossible to observe and record all prevention practices used by respondents due to logistics issues and time consequences. Another limitation was that this study was a cross-sectional study and may not be a good representative of the Selangor population. There was a logistics limitation due to the working activities of wet market workers and their business hours.

## 5. Conclusions

The evidence from this study suggests that there is a need to revise current approaches in delivering information regarding the prevention of specific infectious diseases like leptospirosis in communities. Presence of a significant proportion of non-Malaysian workers in wet market settings has led to new landscape variations in the heterogeneity of social strata in Malaysian population. With a significant number of foreign workers in wet market setting, our healthcare approach is recommended to address specific information delivery in respective languages to prevent the occurrence of specific diseases. These findings enhanced the understanding of delivering health information related to leptospirosis, and the focus should be prioritised not only to Malaysian but also to non-Malaysian workers.

Effective communication of leptospirosis prevention plays an important role in behaviour and attitude concerning leptospirosis. Besides, it is crucial to target the public, especially those with low education levels to have positive attitudes toward leptospirosis prevention. This information can be used to develop relevant interventions aimed at urban markets, especially at high rodent infestation areas. A future study to investigate knowledge, attitude and practice on leptospirosis prevention among wet market workers in rural areas is recommended to provide additional information on the actual view of leptospirosis perception in market workers as a whole.

## Figures and Tables

**Table 1 ijerph-17-01346-t001:** Distribution of socio-demographic among wet market workers.

Variable	Study Group *n* = 147	
	Malaysian Workers *n* = 68	Non-Malaysian Workers *n* = 79
	*n* (%)	*n* (%)
**Age**		
<40 years old	29 (42.6)	57 (72.2)
≥40 years old	39 (57.4)	22 (27.8)
**Gender**		
Male	43 (63.2)	62 (78.5)
Female	25 (36.8)	17 (21.5)
**Education Level**		
Formal Education	60 (88.2)	50 (63.3)
No Formal Education	8 (11.8)	29 (36.7)
**Monthly Income (RM)**		
<RM3000	57 (83.8)	77 (97.5)
≥RM3000	11 (16.2)	2 (2.5)
**Type of business**		
Retail	25 (36.8)	16 (20.3)
Fish, Seafood	4(5.9)	20 (25.3)
Poultry, Livestock	12 (17.6)	7 (8.9)
Vegetable, Fruit	21 (30.8)	25 (31.6)
Food	3 (4.4)	8 (10.1)
Others	3 (4.4)	3 (3.8)

**Table 2 ijerph-17-01346-t002:** Description of knowledge items among market workers.

No	Item	Malaysian (Total *n*: 68) *n* (%) Who Answered Correctly	Non-Malaysian (Total *n*: 79) *n* (%) Who Answered Correctly	*p*
	**Causative Agent**			
1	Leptospirosis caused by bacteria	46 (67.6)	11 (13.9)	0.001
2	Leptospirosis is a zoonotic disease	22 (32.4)	6 (7.6)	0.001
	**Modes of Transmission**			
3	Have cuts and wound on the body	25 (36.8)	8 (10.1)	0.001
4	Mosquitoes bites	33 (48.5)	5 (6.3)	0.001
5	Eating contaminated food	44 (64.7)	11 (13.9)	0.001
	**Signs, symptoms and complication**			
6	Muscle pain	34 (50.0)	13 (16.5)	0.001
7	Yellowing of eyes and skin	20 (29.4)	10 (12.7)	0.1
8	No symptoms	20 (29.4)	6 (7.6)	0.005
9	Lung and kidney failure	37 (54.4)	6 (7.6)	0.001
10	Death	59 (73.5)	11 (13.9)	0.001
	**Risk factors**			
11	Participate in recreational activities	38 (55.9)	11 (13.9)	0.001
12	Cleaning outside house and drain	32 (47.1)	7 (8.9)	0.001
13	Eating street food	41 (60.3)	10 (12.7)	0.001
14	Live near flood area	41 (60.3)	6 (7.6)	0.001
	**Treatment, diagnosis and prevention**			
15	Treat by antibiotic	29 (42.6)	7 (8.9)	0.001
16	Blood screening	45 (66.2)	8 (10.1)	0.001
17	Prevent by vaccination	17 (25.0)	3 (3.8)	0.001
18	Cleanliness in house/workplace area should be prioritized	54 (79.4)	8 (10.1)	0.001
19	Personal hygiene should be prioritized	54 (79.4)	10 (12.7)	0.001
20	Proper storing food/goods to avoid contamination	42 (61.8)	8 (10.1)	0.001
21	Avoid walking without shoes	47 (69.1)	9 (11.4)	0.001
22	Install rat trap	39 (57.4)	8 (10.1)	0.001

**Table 3 ijerph-17-01346-t003:** Description of attitude items among market workers.

No	Variable	Study Group *n* = 147				
		Strongly Agree (%)	Agree (%)	Neutral (%)	Disagree (%)	Strongly Disagree (%)
Q1	Food contaminated with rat excretion are not dangerous to eat.					
	Malaysian	5 (7.4)	3 (4.4)	2 (2.9)	30 (44.1)	28 (41.2)
	Non-Malaysian	4 (5.1)	6 (7.6)	18 (22.8)	36 (45.6)	15 (19.0)
Q2	Presence of rats in house/workplace may cause leptospirosis.					
	Malaysian	39 (57.4)	23 (33.8)	2 (2.9)	3 (4.4)	1 (1.5)
	Non-Malaysian	27 (34.2)	23 (29.1)	22 (27.8)	7 (8.9)	-
Q3	Uncovered dustbin may attract rats to the area.					
	Malaysian	27 (39.7)	24 (35.3)	8 (11.8)	5 (7.4)	4 (5.9)
	Non-Malaysian	25 (31.6)	22(27.8)	27 (34.2)	5 (6.3)	-
Q4	Wading in the flood does not pose a risk of infection					
	Malaysian	7 (10.3)	13 (19.1)	14 (20.6)	22 (32.4)	12(17.6)
	Non-Malaysian	3 (3.8)	9 (11.4)	32 (40.5)	21 (26.6)	14 (17.7)
Q5	Your occupation may expose you to leptospirosis.					
	Malaysian	19 (27.9)	19 (27.9)	15 (22.1)	11 (16.2)	4 (5.9)
	Non-Malaysian	13 (16.5)	12 (15.2)	37(46.8)	16 (20.3)	1(1.3)
Q6	Your hobby/outdoor activity may cause you to get leptospirosis easily.					
	Malaysian	16 (23.5)	24 (35.3)	17 (25.0)	9 (13.2)	2 (2.9)
	Non-Malaysian	8 (10.1)	11 (13.9)	40 (50.6)	18 (22.8)	2 (2.5)
Q7	Unclean environment makes you more prone to have leptospirosis.					
	Malaysian	28 (41.2)	30 (44.1)	9 (13.2)	1 (4.5)	-
	Non-Malaysian	26 (32.9)	18 (22.8)	31 (39.2)	4 (5.1)	-
Q8	Immediate treatment for leptospirosis may avoid more serious complication.					
	Malaysian	34 (50.0)	27 (39.7)	4 (5.9)	3 (4.4)	-
	Non-Malaysian	33 (41.8)	16 (20.3)	24 (30.4)	5 (6.3)	1 (1.3)
Q9	Delayed for leptospirosis treatment may cause death.					
	Malaysian	37 (54.4)	25 (36.8)	4 (5.9)	1 (1.5)	1 (1.5)
	Non-Malaysian	29 (36.7)	19 (24.1)	26 (32.9)	4 (5.1)	1 (1.3)
Q10	Leptospirosis may cause organ complications.					
	Malaysian	25 (36.8)	25 (36.8)	17 (25.0)	-	1 (1.5)
	Non-Malaysian	25 (31.6)	12 (15.2)	34 (43.0)	5 (6.3)	3 (3.8)
Q11	Know about leptospirosis may help in prevention of the disease.					
	Malaysian	32 (47.1)	28 (41.2)	8 (11.8)	-	-
	Non-Malaysian	21 (26.6)	23 (29.1)	27 (34.2)	8 (10.1)	-
Q12	Wearing personal protective equipment (PPE) during cleaning activity is one of disease prevention.					
	Malaysian	30 (44.1)	25 (36.8)	10 (14.7)	3 (4.4)	-
	Non-Malaysian	24 (30.4)	26 (32.9)	24 (30.4)	5 (6.3)	-
Q13	Cleaning program organised by the local authorities enable leptospirosis prevention.					
	Malaysian	25 (36.8)	31 (45.6)	6 (8.8)	4 (5.9)	2 (2.9)
	Non-Malaysian	23 (29.1)	26 (32.9)	26 (32.9)	3 (2.1)	1 (1.3)

**Table 4 ijerph-17-01346-t004:** Distribution of preventive practice items among market workers.

No	Preventive Practice	Study Group *n* = 147				
		Malaysian workers		Non-Malaysian workers		*p*
		Yes	No	Yes	No	
		*n* (%)	*n* (%)	*n* (%)	*n* (%)	
	**Personal Hygiene**					
1	Washing hands with water and soap before and after using toilet	66 (97.1)	4 (5.9)	75 (94.9)	4 (5.1)	0.512 ^1^
2	Washing hands with water and soap before and after preparing food/work	64 (94.1)	4 (5.9)	75 (94.9)	4 (5.1)	0.827 ^1^
	**Environmental hygiene**					
3	Washing equipment used before and after the trade	55 (80.9)	13 (19.1)	64 (81.0)	15 (19.0)	0.984
4	Washing business site	62 (91.2)	6 (8.8)	69 (87.3)	10 (12.7)	0.818
5	Throwing trash into bins provided	64 (94.1)	4 (5.9)	71 (89.9)	8 (10.1)	0.349
	**Specific Protection**					
6	Wearing shoes/boot	52 (76.5)	16 (23.5)	43 (54.4)	36 (45.6)	0.005 *
7	Wearing apron	39 (57.4)	29 (42.6)	26 (32.9)	53 (67.1)	0.003 *
8	Cover each wound using a plaster neatly	58 (85.3)	10 (14.7)	49 (62.0)	30 (38.0)	0.002 *
	**Isolation**					
9	Storing stuff at the end of business inside sealed containers to prevent contamination of rats at night	44 (64.7)	24 (35.3)	45 (57.0)	34 (43.0)	0.338
	**Eradication at source**					
10	Using rat poison to reduce population of rats in the market.	28 (41.2)	40 (58.8)	18 (22.8)	61 (77.2)	0.016 *
11	Using rat trap to reduce population of rats in the market	33 (48.5)	35 (51.5)	15 (19.0)	64 (81.0)	0.0001 *
12	Cleaning at the market site with cooperation from market workers and Malaysian authorities	48 (70.6)	20 (29.4)	38 (48.1)	42 (51.9)	0.006 *

^1^ Fischer Exact Test, * Significant at *p* = 0.05, *n* = 147.

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
