# Peer review of "Awareness, Knowledge, Attitude and Preventive Practice of Leptospirosis Among Healthy Malaysian and Non-Malaysian Wet Market Workers in Selected Urban Areas in Selangor, Malaysia"

_ijerph, 2020, doi:10.3390/ijerph17041346_

Round 1

Reviewer 1 Report

Revision of manuscript ijerph-678672

Dear Authors,

Your manuscript entitled “Awareness, Knowledge, Attitude and Preventive Practice of Leptospirosis among Healthy Malaysian and Non-Malaysian Wet Market Workers in selected urban areas in Selangor, Malaysia” reports a survey, conducted on common people working in wet market, to obtain information about the knowledge on different aspects of Leptospirosis.

The aim is interesting and the work would provide very useful results. However, in my opinion, in the present form it could not be accepted. Most sentences or paragraphs should be rewritten, because they are very difficult to understand. Some examples of this:

Among those animals which are most likely to spread the disease include rodents (20%), marsupials (35%) and bats (35%) [11]. For Malaysian workers, majority of the respondents were those who were more than 40 years old (57.4%), male (63.2%), attained formal education (88.2%) and had income less than RM3000 (83.8%). Most of them involved in retail business and selling vegetables and fruit (64.7%) (Table 1). Majority Malaysian workers also aware that participate in recreational activities (55.9%), eating street food (60.3%) and live near flood area (60.3%) are risk factors for leptospirosis while most of them also know that for prevention of the disease. (Is this sentence incomplete?) There are 13 statements in attitude items which stated either positive attitude or negative attitude towards prevention of leptospirosis. About one third of them also were found to be undecided whether early treatment is important to prevent death and serious complications; Majority of the workers especially non-Malaysian workers was revealed that 82.3% of them do not know about leptospirosis compared to Malaysian workers. Besides, most of the Malaysian workers correctly answered on leptospirosis can cause death and they were able to relate few risk factors such as recreational activity. The percentage markedly low among non-Malaysian workers (less than 20%). (there is no verbs in this sentence).

Note: the sentences reported above are only some examples, but all manuscript has this kind of problems.

The different paragraphs of the introduction should be better linked each others, in particular the first and the third ones.

Authors should try to combine some of the Tables; They are too much and some of them report very few information (for example Table 2). Tables must help the reader to better and immediately understand the results.

In Table 3, point 16: “Detect through blood screening”, probably this sentence is useful for people that must done the survey, but here it would be better to report this like, for example, “serological test” (if I well understand what Authors mean).

“4.1 Distribution of Demographic and Socioeconomic Indicators”, discussion reported in this paragraph should probably be interesting, but, in my opinion, are not necessary and linked with the research.

I sincerely hope that these suggestions will enhance this manuscript. However, if I have made any errors or misinterpretations, I apologize in advance - reviewer

Reviewer 2 Report

The present manuscript shows a study performed on questionnaires to know the preventive practices of leptospirosis of wet market workers in Malaysia

The authors not completely followed the recommended structure with the various section and supplementary data were also included. However, some aspects should be improved, aiming the publication of the manuscript.

Please see below some comments and suggestions, only to further improve the clarity and quality of the manuscript.

Comments:

Pag. 1: revise the cited reference 18 in the text

Pag. 2, line 2: leptospira star with capital

Pag. 4-Pag-5. Material and Methods section is very are to understand, the reading is very difficult and, also, this section doesn’t report in good way the survey. Please revise this section because, for this type of paper, it is the very important part of the work.

Discussion and Conclusion: these sections were wrote not considering the leptospirosis case in human, and also in animal, in Malaysia. Also, there are guidelines or laws for the workers in wet market? The knowledge that workers know leptospirosis but they don’t use any preventive practices are information without consistent because they don’t be related to the humans case or the formation do by instructional programs.

Reviewer 3 Report

Dear Authors: 

Some minor changes are recommended:

Please have an English native speaker to review and edit the text. Please rearrange Table 2 so it is easier for the readers. For table 3, it may be better to present the table in landscape rather than portrait orientation. Alternatively, the "No" column may be removed as it does not really provide important information. Similarly, tables 4 and 5 may be better presented in a landscape orientation. Total Malaysian worker participants were 68 based on several previous tables. However, this 69 answering yes and 2 answering no gave a total of 71. Does this mean there are at least two people answering both Yes and No?  It seems that the total number of Malaysian workers is now 71 rather than 68? At the same time, the non-Malaysian workers has dropped to 76? Is there a reason for such a change, please provide. None of the supplementary tables are listed in the results section. Maybe a good idea to list relevant supplementary tables in the text. Supplementary tables are not part of the downloaded manuscript for review.

Reviewer 4 Report

The purpose of this study is straightforward and well-defined: determine the knowledge of wet workers in Malaysia about leptospirosis and compare Malay vs. non-Malay workers.

Although there is a need for significant improvement in syntax and phrasing, the Results of this study were clear and understandable.  I thought the Discussion section tended to duplicate the Results section too much and could be tidied up.  

Ultimately, this study indicates that there is a significant lack of knowledge and need for an educational campaign in Malaysia that targets immigrants/refugees to help reduce the overall risk of leptospirosis in the country.  

This study serves as a good stimulus for a public policy initiative.  Similar surveys have been performed in other areas, but that doesn't make this one any less relevant.

Round 2

Reviewer 1 Report

Revision of manuscript ijerph-678672

Dear Authors,

Your manuscript entitled “Awareness, Knowledge, Attitude and Preventive Practice of Leptospirosis among Healthy Malaysian and Non-Malaysian Wet Market Workers in selected urban areas in Selangor, Malaysia” reports a survey, conducted on common people working in wet market, to obtain information about the knowledge on different aspects of Leptospirosis.

As previously reported, the aim is interesting and the work would provide very useful results. Authors modified stile and structure of different sentences and paragraphs; this new version of the manuscript resulted more linear and clearer, it is easier to understand and the soundness of the work stand out.

In my opinion, no more modifications are needed.

I sincerely hope that these suggestions will enhance this manuscript. However, if I have made any errors or misinterpretations, I apologize in advance - reviewer

Author Response

dear reviewer

point 1

in my opinion, no more modification needed

response 1

thank you for your valuable opinion

Reviewer 2 Report

The Authors didn't follow any suggestion of first revision. 

Material and methods session is very hard to understand and write in very confusing mode. 

Author Response

Dear reviewer 

Thank you for your opinion, we attached the file to answer your questions.
